# Resistance of the CRISPR-Cas13a Gene-Editing System to Potato Spindle Tuber Viroid Infection in Tomato and *Nicotiana benthamiana*

**DOI:** 10.3390/v16091401

**Published:** 2024-08-31

**Authors:** Ying Wei Khoo, Qingsong Wang, Shangwu Liu, Binhui Zhan, Tengfei Xu, Wenxia Lv, Guangjing Liu, Shifang Li, Zhixiang Zhang

**Affiliations:** 1State Key Laboratory for Biology of Plant Diseases and Insect Pests, Institute of Plant Protection, Chinese Academy of Agricultural Sciences, Beijing 100193, China; khooyingwei@outlook.com (Y.W.K.); wangqingsong@email.swu.edu.cn (Q.W.); binhuizhan@126.com (B.Z.); 2National Citrus Engineering Research Center, Integrative Science Center of Germplasm Creation in Western China (Chongqing) Science City, Citrus Research Institute, Southwest University, Chongqing 400712, China; 3Institute of Industrial Crops, Heilongjiang Academy of Agricultural Sciences, Harbin 150086, China; liushangwu@caas.cn; 4Department of Fruit Science, College of Horticulture, China Agricultural University, Beijing 100193, China; 15833326354@163.com; 5Inner Mongolia Zhongjia Agricultural Biotechnology Co., Ltd., Ulanqab 011800, China; lwx6948288@163.com (W.L.); zjny_liugj@163.com (G.L.)

**Keywords:** tomato, viroid, gene editing, Cas13a, genetic engineering, guide RNAs

## Abstract

Gene-editing technology, specifically the CRISPR-Cas13a system, has shown promise in breeding plants resistant to RNA viruses. This system targets RNA and, theoretically, can also combat RNA-based viroids. To test this, the CRISPR-Cas13a system was introduced into tomato plants via transient expression and into *Nicotiana benthamiana* through transgenic methods, using CRISPR RNAs (crRNAs) targeting the conserved regions of both sense and antisense genomes of potato spindle tuber viroid (PSTVd). In tomato plants, the expression of CRISPR-Cas13a and crRNAs substantially reduced PSTVd accumulation and alleviated disease symptoms. In transgenic *N. benthamiana* plants, the PSTVd levels were lower as compared to wild-type plants. Several effective crRNAs targeting the PSTVd genomic RNA were also identified. These results demonstrate that the CRISPR-Cas13a system can effectively target and combat viroid RNAs, despite their compact structures.

## 1. Introduction

The tomato (*Solanum lycopersicum*) plant is a vital crop in global agriculture with a significant economic impact. However, tomato production faces severe threats from viruses and viroids, including potato spindle tuber viroid (PSTVd) [1,2]. PSTVd is a single-stranded circular, non-encapsidated, non-coding RNA molecule ranging from 356 to 364 nucleotides (nt) [2] and assumes a rod-like secondary structure with five structure–function domains: terminal left (TL), pathogenic (P), central (C), variable (V), and terminal right (TR) domains [3,4]. The viroid replicates within the nuclei via an asymmetric rolling circle mechanism [5]. Briefly, the circular genomic RNA serves as a template for RNA transcription by the host DNA-dependent RNA polymerase II (Pol II). The resulting multimeric antisense RNA is then transcribed into the multimeric sense RNA, forming the replication intermediate, double-stranded RNA. These multimeric sense RNAs are then cleaved into the unit-length sense RNA by certain RNase III and ligated by DNA ligase I, producing mature circular viroid RNA. 

Resistance breeding is the most effective strategy to combat plant viral diseases. However, no natural resistance genes against viroids are available for crop breeding. Thus, genetic engineering has been the primary strategy [6], employing ribonucleases targeting double-stranded RNAs [7,8], catalytic antibodies with ribonuclease activity, antisense and sense viroid RNAs [9,10], catalytic antisense RNAs derived from hammerhead ribozymes [11], hairpin RNAs [12,13,14,15], and artificial small RNAs [16] for RNA interference (RNAi). Despite these efforts, none have been successfully applied in commercial crop breeding due to technological limitations and strict transgene regulation. Gene editing, particularly CRISPR-Cas systems, presents a promising alternative [17].

The CRISPR-Cas13a system can precisely edit RNA molecules within cells, relying on the Cas13a protein with endonuclease activity and a CRISPR RNA (crRNA) that guides the Cas13a protein to bind with the complementary RNA target [18,19,20]. The binding of crRNA with its complementary RNA target changes the Cas13a conformation and activates the protein’s Higher Eukaryotes and Prokaryotes Nucleotide-binding (HEPN) domains, initiating the cleavage of the target RNA [21]. Once the Cas13-crRNA complex binds to its target, the enzyme’s collateral cleavage activity is also triggered. In antiviral strategies, this collateral cleavage allows Cas13a to target not only the intended viral RNA but also multiple other viral RNAs. The potency of this activity has been demonstrated against three distinct ssRNA viruses: lymphocytic choriomeningitis virus (LCMV), influenza A virus (IAV), and vesicular stomatitis virus (VSV). This broad, indiscriminate RNA degradation is crucial for rapidly reducing viral RNA levels, enhancing Cas13a’s effectiveness as an antiviral tool [22]. CRISPR-Cas13a has been successfully employed to enhance plant resistance against RNA viruses [23,24,25,26]. Although its theoretical effectiveness against viroids was recognized, experimental validation was previously lacking. In this study, we demonstrated the decreased levels of viroid RNA using the CRISPR-Cas13a system in both the transient expression system in tomato plants and transgenic system in *Nicotiana benthamiana*. Additionally, we screened crRNAs that efficiently targeted PSTVd genomic RNA, confirming the efficacy of CRISPR-Cas13a against viroid infections.

## 2. Materials and Methods

### 2.1. Construction of pC1300-PSTVd-s

The pBluescript II KS(-)-PSTVd-s plasmid was digested with *Xba* I and *Sal* I, yielding an approximate 0.7 kb fragment. This fragment was excised from the gel and purified using the AxyPrep DNA Gel Extraction Kit (Axygen, Union City, CA, USA), according to the manufacturer’s instructions. The purified fragment was then ligated to *Xba* I- and *Sal* I-digested pCAMBIA1300 binary vector using the LigaFast Rapid DNA ligation system (Promega, Madison, WI, USA). The transformation was performed using *Trelief* 5α chemically competent cells (Tsingke, Beijing, China). Primer sets from Appendix A were used to confirm the successful transformation. The resulting infectious clone was designated as pC1300-PSTVd-s.

### 2.2. Construction of PSTVd-Targeted crRNAs

Candidate crRNAs were chosen in conserved sequences spanning various domains such as TL, P, C, V, and TR. The secondary structure of the sense strand of PSTVd was predicted using the RNAfold [27]. The designed candidate crRNAs were then inserted into the *Bsa* I site within the pCR11 vector to create the corresponding crRNA-Cas13a vectors. The polymerase chain reaction (PCR) condition was performed based on [28]. The successful insertion was confirmed by the polymerase chain reaction (PCR) using the primers set M13F/pCR11-R (Appendix A) and Sanger sequencing. 

### 2.3. Transformation and Agrobacterium Suspension Preparation

Plasmid pC1300-PSTVd-s and crRNA-Cas13a were transformed into *GV3101* competent cells and detected by PCR using primer set SK primer/PSTVd-R and M13F/pCR11-R. The transformation and streak plate method were performed as described previously [29]. A single colony of *Agrobacterium* was transferred to LB. The bacterial cells were pelleted and resuspended in inoculation solution [MES (100 mM), Acetosyringone (200 mM), and Magnesium chloride (100 mM)] to achieve a final OD600 of 1.0. After a resting period of 2 h at room temperature in the dark, the bacterial suspension was used for infiltration [29].

### 2.4. Transient Expression of Recombinant Vectors in ‘Rutgers’ Tomato Plants

In the proximal region of the leaf, each crRNA-Cas13a and empty vector (EV) was transiently expressed in 10-day-old ‘Rutgers’ tomato plants through agroinoculation. After two days, the distal region of the leaf was inoculated with PSTVd-s. Each treatment included six biological replicates and three technical replicates. Healthy mock controls were inoculated with an empty pCAMBIA1300 vector. The plants were transformed with syringe inoculation. Using a 1 mL syringe, without a needle, applying gentle pressure to the underside of the seedling leaf, while exerting counter-pressure with a finger on the opposite side, facilitated successful infiltration, as indicated by a spreading “wet” area on the leaf.

### 2.5. Preparation of DIG-Labeled Probe and Northern blot Hybridization

A DIG-labeled PSTVd RNA probe was prepared by in vitro transcription using T7 RNA polymerase (NEB, Ipswich, MA, USA) from the linearized recombinant plasmid of pGEM-PSTVd [30].

Total nucleic acid was extracted using TransZol (Transgene Biotech, Beijing, China) and analyzed by northern blot hybridization [30]. Briefly, total RNA was separated on a denaturing formaldehyde agarose gel and transferred to a nylon membrane using a vacuum blotter (BIO-RAD, Hercules, CA, USA). The transferred RNA was cross-linked with the membrane using a UV crosslinker (CL-1000 UVP, Analytik Jena US LLC, CA, USA). Prehybridization and hybridization were performed at 68 °C. Then, the membrane was washed by washing buffers, blocked using a blocking reagent, incubated with an anti-Digoxigenin-AP antibody (Roche Diagnostics GmBH, Mannheim, Germany), and finally added to CSPD to generate fluorescence. The signal was collected using the Tanon 5200 chemiluminescent imaging system (Tanon, Shanghai, China).

### 2.6. Development and Selection of Transgenic N. benthamiana Plants

Transgenic *N. benthamiana* plants expressing Cas13a protein and crRNA of CCR2(+) were generated as described previously [31]. T1 transgenic lines were selected using hygromycin (20 mg/L). T2 seedlings were screened by the PCR (Appendix A). The expression levels of Cas13a in T2 transgenic lines were quantified using reverse-transcription quantitative real-time PCR (RT-qPCR) (Appendix A). The lines with higher expression levels were selected for the following inoculation assay (see Section 2.7).

### 2.7. PSTVd Inoculation in Transgenic N. benthamiana CCR2(+) Plants

Transgenic *N. benthamiana* plants expressing the CCR2(+) construct were utilized to evaluate the viroid resistance provided by the CRISPR-Cas13a system. The selected T2 transgenic lines 2 and 6 were inoculated with PSTVd through agroinfiltration. The symptoms were monitored and compared among the infected transgenic CCR2(+) lines, EV controls, and mock-inoculated healthy controls.

### 2.8. Densitometry and Statistical Analysis

After detecting the hybridized signals on the northern blot, we quantified the band intensities of both the PSTVd and 18S rRNA load controls using the GIS Tanon system. The normalized value was analyzed with a *t*-test and one-way ANOVA in the GraphPad Prism 9.3.0 software. The *t*-test assessed differences in relative PSTVd accumulation between tomato plants under EV and CCR1(+) conditions. One-way ANOVA evaluated the differences among tomato plants subjected to EV, TL(+), TR(+), CCR2(+), TL(−), CCR1(−), TR(−), and CCR2(−) in the transient expression experiment, as well as among transgenic *N. benthamiana* Line 2, Line 6, and EV. *Protein Phosphatase 2A* (*PP2A*) was used as the reference gene for the normalization of Cas13a expression levels.

## 3. Results

### 3.1. The CRISPR-Cas13a System can Efficiently Reduce Viroid Accumulation

To establish that gene-editing technology can confer resistance to viroid infection in plants, Cas13a was transiently expressed in the tomato plants along with crRNA complementary to the upper central conserved region (CCR) of PSTVd. Tomato plants expressing both Cas13a and crRNA [CCR1(+)], as well as control plants expressing Cas13a (EV) (18 plants per treatment), were agroinfiltrated with an infectious clone of the severe PSTVd variant (pC1300-PSTVd-s). PSTVd accumulation levels were assessed by northern blot hybridization 30 days post-infiltration (dpi). As shown in Figure 1A,B, the CCR1(+) plants exhibited significantly lower PSTVd RNA levels compared to the EV plants, with a reduction of more than 70%. These findings demonstrate that the CRISPR-Cas13a system effectively targets viroid RNA in tomato plants.

### 3.2. crRNAs Targeting PSTVd Genomic RNA

The efficiency of the CRISPR-Cas13a system is influenced by the sequences and structure of the regions targeted by crRNAs [21]. For broad-spectrum resistance and to disrupt critical biological functions, crRNAs should target conserved sequences in the viroid RNA. We designed various 23-nucleotide (nt) crRNAs targeting both the sense and antisense strands of the PSTVd genome. These targets include terminal conserved regions of TL, upper and lower strands of CCR related to viroid replication [32] and TR, which contains structural motifs essential for viroid trafficking [33] (Figure 2). BLAST analysis confirmed that the designed crRNAs had no on-target sites in the genomes of both the tomato (GCF_000188115.5, SL3.1) and *N. benthamiana* (GCA_034376525.1, ASM3437652v1).

### 3.3. Screen of crRNAs with High Efficiency Targeting PSTVd RNA

The constructs pC1300-PSTVd-s and crRNA-Cas13a were successfully transformed into the *Agrobacterium tumefaciens* strain GV3101, as confirmed by the PCR at approximately 700 bp and 400 bp (Appendix A). To evaluate the effectiveness of the designed crRNAs in targeting PSTVd RNA, we used *Agrobacterium*-mediated transient expression to introduce the CRISPR-Cas13a system into the tomato plants. Two days post-transformation, the plants were infiltrated with *Agrobacterium* carrying either the EV or PSTVd infectious clone (pC1300-PSTVd-s) via syringe inoculation. Each treatment group consisted of 18 plants, and the experiment was conducted in triplicate. At 30 days post-inoculation (dpi), no symptoms were observed in the mock-inoculated healthy control plants. In contrast, all EV plants exhibited severe stunting, leaf malformation, and vein necrosis (Figure 3A,B). When CRISPR-Cas13a and crRNAs were expressed, the severity of symptoms varied depending on the specific crRNA constructs. Plants expressing crRNAs targeting the sense strand of PSTVd, as well as those targeting the antisense TR(−) and CCR2(−) regions, showed minimal leaf malformation and no significant stunting. Plants expressing TL(−) and CCR1(−) targeting the antisense strand displayed noticeable stunting and leaf malformation but were less severe than the symptoms in EV plants (Figure 3B).

Northern blot hybridization was used to analyze the PSTVd accumulation levels in these tomato plants (Figure 3C,D). Compared to the EV plants, PSTVd genomic RNA was undetectable in plants expressing TL(+) and CCR2(+), significantly reduced in plants expressing CCR1(+), TR(−), and CCR2(−), and at intermediate levels in TR(+) and CCR1(−) plants. Only plants expressing TL(−) accumulated PSTVd genomic RNA at levels comparable to the EV plants (Figure 3 and Appendix A). Quantification of the northern blot results showed that all crRNA-Cas13a constructs, except TL(−), significantly inhibited PSTVd accumulation (F[7, 262] = 251, *p* < 0.0001) (Figure 3E).

### 3.4. Stable Expression of the CRISPR-Cas13a System Attenuates Viroid Accumulation

To assess the viroid resistance efficacy of the CRISPR-Cas13a system in stably transformed plants, we expressed the CCR2(+) construct in *N. benthamiana* plants through transgenesis. Five transgenic *N. benthamiana* lines (L1, L2, L6, L8, and L11) were confirmed to contain the CCR2(+) insert in the T2 generation via PCR (Appendix A). The T2 transgenic lines used in the experiments did not show phenotypic alterations with respect to wild-type plants (Appendix A). Lines 2 and 6 were selected for further experiments due to their higher accumulation of Cas13a transcripts (Appendix A). The CCR2(+) construct was chosen because it resulted in the lowest PSTVd accumulation in transiently transformed tomato plants. As a control, an EV was also introduced into the plants. Both CCR2(+) and EV transgenic plants were agroinfiltrated with PSTVd, and PSTVd accumulation was evaluated at 21 dpi using northern blot hybridization. No symptoms were exhibited in either transgenic or wild-type *N. benthamiana* (Figure 4A). In the transgenic lines (L2 and L6), PSTVd accumulation was unobservable, whereas it was observable in the EV plants (Figure 4B). These results indicate that the stable expression of the CRISPR-Cas13a system effectively reduces viroid accumulation.

## 4. Discussion

This study determined the inhibition activity of the CRISPR-Cas13a system on viroid RNA, which folds into a compact structure due to extensive intramolecular base pairs. This extends the application of gene-editing technology to RNAs with complex structures that may be resistant to RNAi-mediated degradation [34]. Additionally, we demonstrated that the gene-editing system can confer resistance against viroid infection, similar to its effectiveness against plant RNA virus infections [17] in both transient and transgenic expression systems. This provides a new strategy for viroid control.

Northern blot hybridization revealed varying levels of PSTVd accumulation, ranging from undetectable to high, depending on the specific crRNAs used. This underscores the need to design and optimize crRNAs when employing the CRISPR/Cas13a system for viroid resistance. In transgenic *N. benthamiana* plants expressing the optimized crRNA, PSTVd RNA was barely detectable, suggesting an enhanced resistance to viroid infection.

Our findings showed that crRNAs such as TR(−) and CCR2(−) could effectively guide Cas13a to reduce the antisense genomic RNA of PSTVd (Figure 3. The viroid antisense strand, synthesized in the nucleus during replication [35], usually exists as a double-stranded RNA intermediate [36,37]. However, Cas13a prefers targeting single-stranded RNA [18], indicating that the nascent antisense strand of viroid should be targeted by Cas13a in the nucleus. The designed crRNAs targeting different positions on the sense and antisense PSTVd genomic RNA showed varied effects on the PSTVd accumulation levels, suggesting differences in crRNA affinity, sequence, or target structure influencing Cas13a’s cleavage activity. crRNAs targeting the PSTVd sense strand appeared more efficient than those targeting the antisense strand, possibly due to the dual nucleus and cytoplasm localization of the viroid sense strand, increasing its probability of being cleaved by Cas13a.

Future research should aim to elucidate the molecular mechanisms underlying Cas13a-mediated degradation of viroid RNA to optimize crRNA design for viroid RNA targeting. Studies should explore the long-term effects and dynamics of viroid infection in transgenic plants beyond 21 dpi. While RNAi is a powerful tool for controlling plant RNA viruses [38,39], its efficiency can be compromised by off-target gene silencing, environmental factors, delivery challenges, and potential viral resistance [26,40,41]. CRISPR-Cas13, targeting RNA directly with crRNAs, offers higher specificity and rapid viral RNA degradation, enhancing its effectiveness against diverse strains [21,42,43]. The precision and robustness of CRISPR-Cas13 make it a superior option for viroid control compared to RNAi. Although no natural resistance gene has been identified for viroids, a tolerance to PSTVd in its wild species is a dominant trait that can be introduced into cultivated varieties by crossing [44]. Once resistance or susceptibility genes for viroid infection are identified, they can be precisely edited using gene-editing technology for viroid control. Until then, directly targeting viroid RNA with crRNAs remains a suitable strategy.

## 5. Conclusions

This study highlights the promising potential of the CRISPR-Cas13a system in imparting resistance against PSTVd in tomato plants and *N. benthamiana*, using both transient and transgenic approaches. The alleviation of PSTVd-induced symptoms and substantial reduction in viroid accumulation in the treated plants emphasize the effectiveness of this strategy. Overall, our findings underscore the efficacy of CRISPR-based strategies for precise and efficient viroid control, promising enhanced plant health and agricultural productivity.

## Figures and Tables

**Figure 1 viruses-16-01401-f001:**
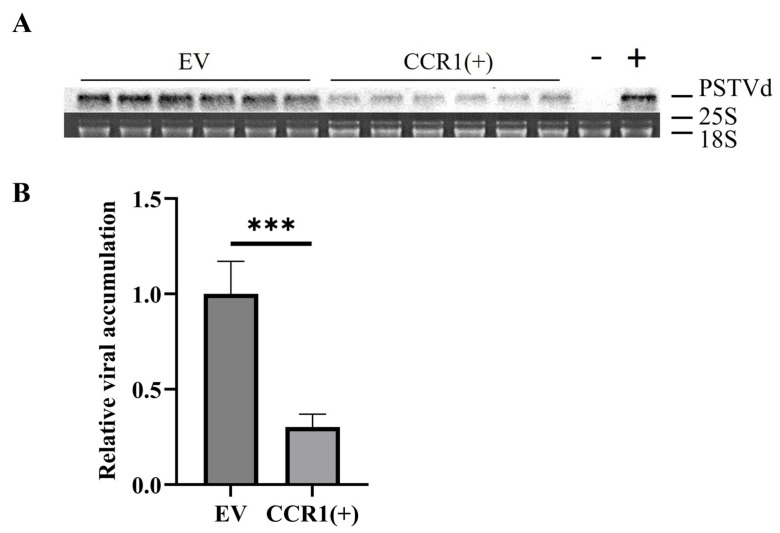
CRISPR-Cas13a effectively inhibits viroid RNA accumulation. (**A**) Northern blot hybridization analysis of tomato plants transiently expressing CRISPR-Cas13a targeting the upper central conserved region of PSTVd [CCR1(+)] compared to the controls (EV), using rRNA as a loading control. The “− “ symbol indicates healthy mock controls and the “+” symbol indicates the PSTVd-positive control. (**B**) Quantitative analysis (n = 18) of Northern blot results. Error bars represent standard deviation (SD). *** indicates a statistically significant difference compared to the control group (*p* < 0.001).

**Figure 2 viruses-16-01401-f002:**
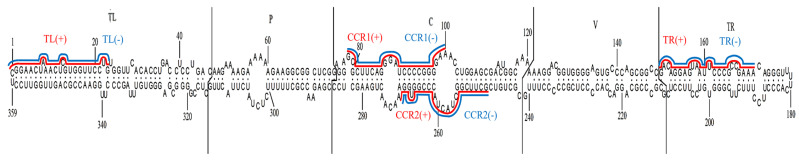
Secondary structure of PSTVd genomic RNA (Genbank accession no. MK303581). Designed crRNAs targeting sense and antisense strands are highlighted by red and blue lines, respectively. Structural domains are labeled as TL (terminal left), P (pathogenicity), C (central), V (variable), and TR (terminal right).

**Figure 3 viruses-16-01401-f003:**
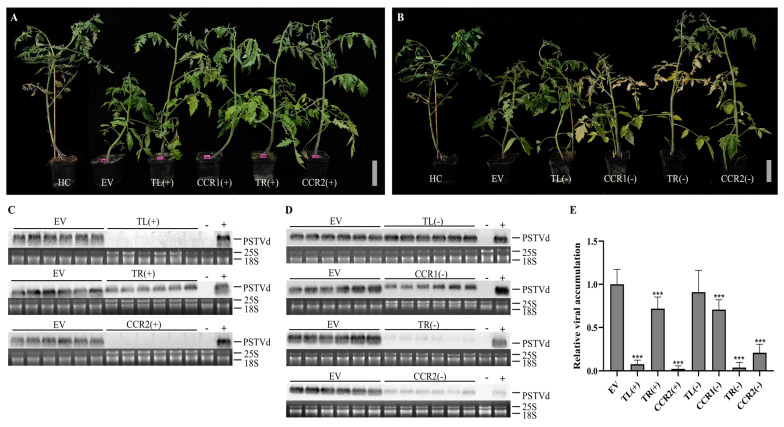
Screening of effective crRNAs targeting PSTVd genomic RNA. (**A**,**B**) Symptoms of tomato plants expressing different crRNA-Cas13a constructs at 30 days post-inoculation. HC represents the healthy control. Plants expressing crRNAs targeting the sense strand of PSTVd, as well as the antisense TR(−) and CCR2(−) regions, exhibited minimal leaf malformation and no significant stunting. In contrast, the plants expressing TL(−) and CCR1(−), targeting the antisense strand, displayed noticeable stunting and leaf malformation. Scale bar = 10 cm. (**C**,**D**) Northern blot hybridization of the plants shown in A and B using rRNA as loading control. (**E**) Quantitative analysis (n = 18) of Northern blot results from (**C**,**D**). Error bars represent standard deviation (SD). No bands were detected in the plants expressing TL(+) and CCR2(+), while the plants expressing TR(−) and CCR2(−) showed significantly reduced PSTVd levels. CCR1(−) and TR(+) plants exhibited intermediate levels of PSTVd accumulation. TL(−) was the only construct that allowed PSTVd RNA accumulation at levels comparable to the EV plants. Quantification revealed that all crRNA-Cas13a constructs, except TL(−), significantly inhibited PSTVd accumulation. *** indicates a statistically significant difference compared to the control group (*p* < 0.001).

**Figure 4 viruses-16-01401-f004:**
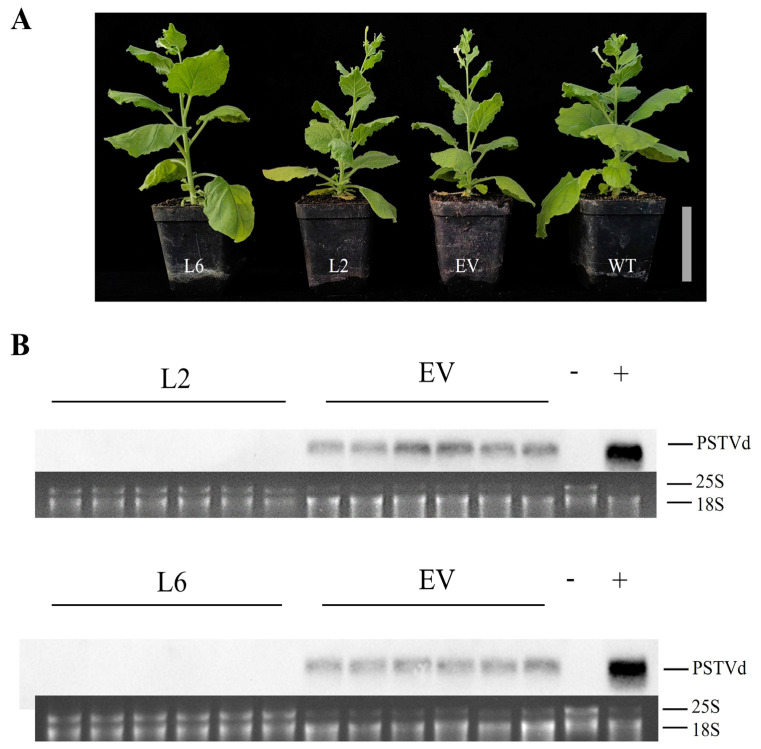
Stable expression of the CRISPR-Cas13 system effectively reduces viroid accumulation in transgenic *Nicotiana benthamiana*. (**A**) No symptoms were observed in either transgenic *N. benthamiana* or wild-type plants at 21 days post-PSTVd inoculation. (**B**) PSTVd accumulation was assessed in L2, L6, and EV plants using Northern blot hybridization, with 18S rRNA as the loading control. Scale bar = 10 cm. Data represent six biological replicates.

## Data Availability

Data are contained within the article and Appendix A.

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
