# Peer review of "Resistance of the CRISPR-Cas13a Gene-Editing System to Potato Spindle Tuber Viroid Infection in Tomato and Nicotiana benthamiana"

_viruses, 2024, doi:10.3390/v16091401_

Round 1
Reviewer 1 Report
Comments and Suggestions for Authors
Khoo et al describe the use of both a transient and transgenic CRISPR-Cas13a system that reduces the levels of PSTVd during infection.
The report is easy to follow, although the authors may want to do a thorough copyedit before resubmitting (e.g CRISSPR line 21).
The results of the work are worth publication but for me the contents and descriptions are a lightweight and skim over essential details. A much better description of the transgenics and characterization of these plants in the absence and presence of the viroid is required before publication in this journal. The legends for the figures too are not standalone and need more information. Northern blot analysis of the RNA levels is good but should be supported by a more sensitive method like qRT-PCR.
60-65 – give more details on the CRISPR-Cas13a system to guide the uninitiated and maybe explain why conceptually this will work: Cas13a once initiated specifically by the target will cleave collaterally all RNAs. Why is that an advantage?
87-92 – there seems to be too much detail on a straightforward technique here that could be described with a simple citation. It’s fine to add details but there’s an imbalance here in that some methods and results that are important are practically ignored.
97 – check use of acropetal – meaning “proceeding from the base toward the apex or from below upward”.
105-115 – provide more details of Northern – cross-linker, probe, DIG labelling.
121 - give full meaning to acronyms EV and CCR
124 – there is no mention of the methods to produce transgenic plants and how they were selected and characterized. Please include.
160 – post-transformation. How were the plants transformed?
163 – symptoms
174 – “undetectable” – and by qRT-PCR?
182 – is there any difference between A & B or are they just representative?
189 - N. benthamiana plants through
193 – why only at 21 dpi? Why not an extended period too?
199 – what do the plants look like? I’d be interested to know the off-target effects of expressing Cas13a.
Comments on the Quality of English Language
The report is easy to follow, although the authors may want to do a thorough copyedit before resubmitting (e.g CRISSPR line 21).
Author Response
Thank you very much for reviewing our manuscript. The following are our responses to your comments and corresponding revisions were highlighted in red in revised manuscript.
Comments 1: The report is easy to follow, although the authors may want to do a thorough copyedit before resubmitting (e.g CRISSPR line 21).
Response: The typographical error "CRISSPR" in line 22 has been corrected to "CRISPR".
Comments 2: The results of the work are worth publication but for me the contents and descriptions are a lightweight and skim over essential details. A much better description of the transgenics and characterization of these plants in the absence and presence of the viroid is required before publication in this journal. The legends for the figures too are not standalone and need more information. Northern blot analysis of the RNA levels is good but should be supported by a more sensitive method like qRT-PCR.
Response: We appreciate your recognition of the significance of our work and your suggestions. We would like to address your concerns as followings:
We have expanded the section on the creation and characterization of transgenic plants, providing more detailed information on methods and phenotypic characterizations of viroid-infected and non-infected plants in line 128-141, Figure 4A and Figure S3.
The legends of all figures have been revised to ensure they are standalone including a comprehensive explanation of the experimental setup, methods, and the significance of the results.
We appreciate the suggestion to use more sensitive qRT-PCR to analyze RNA levels, we also believe that qRT-PCR analysis could further support our conclusion. Considering that the results of northern blotting analysis are reliable and robust, and are sufficient to support our conclusion. They can obviously show the differences in viroid accumulation between different treatments. Thus, the absence of precise analysis of qRT-PCR does not weak the conclusion that expression of CRISPR/Cas13a and crRNA inhibits PSTVd accumulation in tomatoes. In addition, the cost limit is another reason.
We have provided detailed descriptions for plant transformation, viroid inoculation, RNA extraction, and northern blotting, including specific details on the reagents used and the experiment conditions.
Comments 3: 60-65 – give more details on the CRISPR-Cas13a system to guide the uninitiated and maybe explain why conceptually this will work: Cas13a once initiated specifically by the target will cleave collaterally all RNAs. Why is that an advantage?
Response: Thank you for the suggestion. We have added information about the CRISPR-Cas13a system as followings: “The CRISPR-Cas13a system can precisely edit RNA molecules within cells, relying on Cas13a protein with endonuclease activity and a CRISPR RNA (crRNA) that guides Cas13a protein to bind complementary RNA target [18-20]. The bind of crRNA with its complementary RNA target changes Cas13a conformation and activates the protein’s Higher Eukaryotes and Prokaryotes Nucleotide-binding (HEPN) domains, initiating the cleavage of the target RNA [21]. Once the Cas13-crRNA complex binds to its target, the enzyme's collateral cleavage activity is also triggered. This causes Cas13a to cleave not only the perfectly matched target RNA but also other non-target RNAs, resulting in a broad reduction in the over-all viral RNA population. The effectiveness of this collateral activity has been demon-strated by Cas13a's ability to identify and degrade active target sites across the LCMV (Lymphocytic Choriomeningitis Virus) genome, disrupting the virus's ability to repli-cate and carry out its life cycle within the host cell [22]. This broad and indiscriminate RNA degradation is crucial for rapidly reducing viral RNA levels, making Cas13a an effective tool in antiviral strategies.
Comments 4: 87-92 – there seems to be too much detail on a straightforward technique here that could be described with a simple citation. It’s fine to add details but there’s an imbalance here in that some methods and results that are important are practically ignored.
Response: Thank you for the feedback. We have streamlined the related descriptions by providing a citation.
Comments 5: 97 – check use of acropetal – meaning “proceeding from the base toward the apex or from below upward”.
Response: We have replaced the “acropetal” and “basipetal” with “proximal” and “distal”, respectively in line 105,107.
Comments 6: 114-126 – provide more details of Northern – cross-linker, probe, DIG labelling.
Response: We briefly introduced the method as follows:
A DIG-labeled PSTVd RNA probe was prepared by in vitro transcription using T7 RNA polymerase (NEB, USA) from the linearized recombinant plasmid of pGEM-PSTVd [30].
Total nucleic acid was extracted using TransZol (Transgene Biotech, China) and analyzed by northern blot hybridization [30]. Briefly, total RNA was separated on a denaturing formaldehyde agarose gel and transferred to a nylon membrane using a vacuum blotter (BIO-RAD, USA). The transferred RNA was cross-linked with the membrane using a UV crosslinker (CL-1000 UVP, USA). Prehybridization were per-formed at 68°C. Then, the membrane was washed by washing buffers, blocked using blocking reagent, incubated with anti-Digoxigenin-AP antibody (Roche Diagnostics GmBH, Germany), and finally added CSPD to generate fluorescence. The signal was collected using the Tanon 5200 chemiluminescent imaging system (Tanon, China).
Comments 7: 121 - give full meaning to acronyms EV and CCR
Response: "EV" and "CCR", representing ‘empty vector’ and ‘central conserved region’.
Comments 8: 124 – there is no mention of the methods to produce transgenic plants and how they were selected and characterized. Please include.
Response: Thank you for pointing out this problem. We have included a detailed description of the methodology in line 128-134 in the revised manuscript. The added text is as follows: “
2.6 Development and selection of transgenic N. benthamiana plants
Transgenic N. benthamiana plants expressing Cas13a protein and crRNA of CCR2(+) were generated as described previously [31]. T1 transgenic lines were selected using hygromycin (20 mg/L). T2 seedlings were screened by PCR (Table S2). The expression levels of Cas13a in T2 transgenic lines were quantified using reverse-transcription quantitative real-time PCR (RT-qPCR) (Table S2). Lines with higher expression levels were selected for the following inoculation assay (see 2.7).
Comments 9: 160 – post-transformation. How were the plants transformed?
Response: The plants were transformed using syringe infiltration. A 1 mL syringe (with the needle removed) was used to gently apply the Agrobacterium suspension to the underside of the seedling leaves. Gentle pressure was applied to the leaf while counter-pressure was exerted on the opposite side, facilitating infiltration, as indicated by the appearance of a spreading “wet” area on the leaf.
Comments 10: 163 – symptoms
Response: It has been corrected in the revised manuscript.
Comments 11: 174 – “undetectable” – and by qRT-PCR?
Response: Here, "undetectable" refers to the results obtained from northern blot analysis. Right, viroid may be able to be detected by more sensitive qRT-PCR. In the future, we will determine the change of viroid accumulations in a more sensitive way.
Comments 12: 182 – is there any difference between A & B or are they just representative?
Response: There is indeed a difference between A and B. Group A consists of plants inoculated with Agrobacterium-mediated transient expression to introduce the CRISPR-Cas13a system targeting the sense strand of PSTVd (Potato spindle tuber viroid). In contrast, Group B includes plants inoculated similarly but with the CRISPR-Cas13a system targeting the antisense strand of PSTVd.
Comments 13: 189 - N. benthamiana plants through
Response: We have added the word "plants" after "N. benthamiana," in line 263
Comments 14: 193 – why only at 21 dpi? Why not an extended period too?
Response: We appreciate your interest in this point and your suggestion. The extended observations are important. We are doing this work, however, we just got some preliminary results and we are repeating them. We have also indicated in the manuscript that future studies will focus on these extended time points to provide a more comprehensive understanding of viroid infection. The findings from these ongoing studies will be presented in subsequent publications, as the current manuscript primarily focuses on the early-stage Responses at 21 dpi.
Comments 15: 199 – what do the plants look like? I’d be interested to know the off-target effects of expressing Cas13a.
Response: Although we did not observe any abnormal phenotype of the plants, we could not exclude possible off-target at the molecular level. We are also interested in the possible off-target effects of expressing Cas13a, which is also one of our future research topics. Relevant figures are included in the supplementary material file (Figure S3) for further examination.

Reviewer 2 Report
Comments and Suggestions for Authors
Overall comment: The manuscript provides evidence of using CRISPR Cas-13 to develop resistance against Potato Spindle tuber viroid (PSTVd) in tomato and Nicotiana benthamiana. The authors provide experiments to demonstrate the effectiveness of CRISPR-Cas13 against PSTVd. However, the authors need to provide more information about some experimental procedures to have the manuscript accepted in the journal. In addition, they need to provide atleast a control experiment as suggested in the detailed comment in supplementary information section below.
Detailed comments:
Title:
- The authors show the development of resistance against only PSTVd by CRISPR-Cas13a in plants. Hence, the title should be more specific and modified to indicate the name of the particular viroid and the plant.
Materials and methods:
2.5. Northern Blot hybridization : Did the authors use a denaturing gel for their Northern hybridization? If yes, they need to provide more details?
2.6 Statistical analysis : The authors need to provide details about load control? In addition, how did they normalize the variation in the load while quantifying their northern blot signals?
Results:
Figure 1, Figure 3, and Figure 4 : Lacks information about load controls both in the legend or within the figure.
Figure 3 D : For CCR2 (-), the positive control seems to have below detectable signals. Can the authors clarify why there is a poor signal for this positive control?
Figure 4 : Why Line 2 and line 6 were selected for this experiment? Was it based on the levels of the crRNA’s or the CRISPR-Cas13 expression? If yes, the authors are encouraged to provide the expression level of the constructs in the transgenic plants. The authors need to show the whole plant pictures of the transgenic (Ev and CRISPR-Cas13a) plants to show there are no phenotypic differences between the plants because of the stable expression of the CRISPR-Cas 13.
Discussion:
The authors do not provide evidence for the cleavage activity of CRISPR-Cas13a on viroid RNA. The results presented in the manuscript suggest that in the presence of CRISPR-Cas13a the viroid RNA accumulates to a lower level than in control plants, but there is no direct evidence to support cleavage of viroid RNA. Hence, the authors must modify the text in lines 202-203 (first two lines of Discussion) accordingly. To show and claim the cleavage activity of CRISPR-Cas 13a on the viroid RNA, the authors must examine the accumulation of viroid RNA in the presence of a mutated CRISPR-Cas13a lacking cleavage activity.
Supplementary information:
The agarose gel pictures provided in the supplementary figure for the northern blots all lack size markers (DNA or RNA ladders), making it hard to identify these bands scientifically. The authors also don’t mention the load controls in the manuscript. The authors are encouraged to provide an agarose gel picture and a corresponding northern blot picture with DNA/RNA size ladders, positive control samples, healthy mock, and negative control samples to help indicate the bands on the agarose gel picture and the band on the northern blot.
Typographical :
Figure 4 ; The author have provided a explanation about error bars in the legend. It seems to be a typographical mistake.
In Lines 192 and 193, the authors state “Both transgenic and EV plants ….. at 21 dpi using northern blot hybridization.” This sentence can be misleading as both CRISPR Cas13 and EV plants are transgenic. The authors need to modify this sentence accordingly.
Author Response
Thank you very much for taking the time to review this manuscript. Please find the detailed responses below, along with the corresponding revisions and corrections highlighted in the re-submitted files. Changes made in response to reviewer 2’s comments are highlighted in blue. Additionally, some of these changes overlap with those requested by reviewer 1, which are highlighted in red.
Comments 1:
Title:- The authors show the development of resistance against only PSTVd by CRISPR-Cas13a in plants. Hence, the title should be more specific and modified to indicate the name of the particular viroid and the plant.
Response: Thank you for your suggestion. We have revised the title to “Resistance of the CRISPR-Cas13a Gene-Editing System to Potato Spindle Tuber Viroid Infection in tomato and Nicotiana benthamiana.”
Comments 2:
Materials and methods:
2.5. Northern Blot hybridization: Did the authors use a denaturing gel for their Northern hybridization? If yes, they need to provide more details?
Response: Yes, we used a denaturing formaldehyde gel for the Northern hybridization. The detailed methodology has been provided in line 124-126 in the revised manuscript.
Comments 3: 2.6 Statistical analysis: The authors need to provide details about load control? In addition, how did they normalize the variation in the load while quantifying their northern blot signals?
Response: We used 18S rRNA as the loading control. The intensity of the PSTVd signal for each sample was divided by the intensity of the 18S rRNA signal, resulting in a normalized ratio that corrected for any differences in RNA loading or transfer. These normalized values were used for statistical analysis, ensuring that the observed differences in PSTVd accumulation levels accurately reflected biological changes rather than technical variability.
Comments 4: Results:
Figure 1, Figure 3, and Figure 4 : Lacks information about load controls both in the legend or within the figure.
Response: We have added detailed information about the loading controls, 18s RNA in these figures and corresponding legends.
Comments 5: Figure 3 D : For CCR2 (-), the positive control seems to have below detectable signals. Can the authors clarify why there is a poor signal for this positive control?
Response: The positive control for CCR2 (-) was indeed a low level. Uneven distribution of viroid in different tissues of PSTVd-infected tomato plants should be the main reason because the RNA of different positive controls was extracted from different parts of plants. The possibility of operation was also not completely excluded. However, the weak signal did not impact the results because all the EV treatments produced strong signals, which ensures the success of inoculation and hybridization.
Comments 6: Figure 4 : Why Line 2 and Line 6 were selected for this experiment? Was it based on the levels of the crRNA’s or the CRISPR-Cas13 expression? If yes, the authors are encouraged to provide the expression level of the constructs in the transgenic plants. The authors need to show the whole plant pictures of the transgenic (Ev and CRISPR-Cas13a) plants to show there are no phenotypic differences between the plants because of the stable expression of the CRISPR-Cas 13.
Response: We thank these insightful comments. Right, the two lines were selected based on the levels of CRISPR-Cas13a expression. We obtained a total of 5 lines with successful expression of crRNA and CRISPR-Cas13a based on PCR detection. The expression levels are determined by RT-qPCR. The results of RT-qPCR were shown in Figure S3C in the revised manuscript. Transgenic plants of all lines have no visible phenotypic differences compared with wild-type plants. The figure of the whole transgenic plants including both empty vector (EV) and CRISPR-Cas13a-expressing plants, was added as Figure S3. and in response to the reviewer’s suggestion, we have now included additional data in the revised manuscript showing the expression levels of CRISPR-Cas13a in these transgenic lines (Figure S3). Furthermore, we understand the reviewer’s concern regarding potential phenotypic differences due to the stable expression of CRISPR-Cas13a. To address this, we have also provided images of the whole transgenic plants, including both empty vector (EV) and CRISPR-Cas13a-expressing plants, in the supplementary material (Figure S3). These images demonstrate that there are no observable phenotypic differences between the transgenic plants, confirming that the stable expression of CRISPR-Cas13a does not affect plant morphology.
Comments 7: Discussion:
The authors do not provide evidence for the cleavage activity of CRISPR-Cas13a on viroid RNA. The results presented in the manuscript suggest that in the presence of CRISPR-Cas13a the viroid RNA accumulates to a lower level than in control plants, but there is no direct evidence to support cleavage of viroid RNA. Hence, the authors must modify the text in lines 202-203 (first two lines of Discussion) accordingly. To show and claim the cleavage activity of CRISPR-Cas 13a on the viroid RNA, the authors must examine the accumulation of viroid RNA in the presence of a mutated CRISPR-Cas13a lacking cleavage activity.
Response: We appreciate the reviewer’s feedback. We completely agree with this. Thus, we have replaced “cleavage’ with “inhibition” in line 248. It is a valuable experiment to examine viroid RNA accumulation in the presence of a mutated CRISPR-Cas13a lacking cleavage activity. We can try this in future research.
Comments 8: Supplementary information:
The agarose gel pictures provided in the supplementary figure for the northern blots all lack size markers (DNA or RNA ladders), making it hard to identify these bands scientifically. The authors also don’t mention the load controls in the manuscript. The authors are encouraged to provide an agarose gel picture and a corresponding northern blot picture with DNA/RNA size ladders, positive control samples, healthy mock, and negative control samples to help indicate the bands on the agarose gel picture and the band on the northern blot.
Response: We appreciate your suggestion. In the revised manuscript, we have mentioned and labeled the load controls accordingly. The suggestion “to provide an agarose gel picture and a corresponding northern blot picture with DNA/RNA size ladders, positive control samples, healthy mock, and negative control samples to help indicate the bands on the agarose gel picture and the band on the northern blot.” is very good and very helpful for our experiment design. Our present figures are not perfect, however, we think they are reliable and sufficient for the comparison between the two treatments. Thus, we prefer to use the origin figures with detailed labels of loading control.
Comments 9: Typographical :
Figure 4 ; The author have provided a explanation about error bars in the legend. It seems to be a typographical mistake.
Response: Thank you. I have corrected this error by removing the “Error bars represent SD.” in the legend.
Comments 10: In Lines 192 and 193, the authors state “Both transgenic and EV plants ….. at 21 dpi using northern blot hybridization.” This sentence can be misleading as both CRISPR Cas13 and EV plants are transgenic. The authors need to modify this sentence accordingly.
Response: We appreciate your observation. To clarify, we revised the sentence to eliminate any ambiguity as follows in line 234-236: "Both CCR2(+) and EV transgenic plants were agroinfiltrated with PSTVd, and PSTVd accumulation was evaluated at 21 days post-infiltration (dpi) using northern blot hybridization.”

Round 2
Reviewer 1 Report
Comments and Suggestions for Authors
Much improved descriptions. Thanks
Two minor comments:
57 – binding
62-63 – “non-target RNAs, resulting in a broad reduction in the overall viral RNA population.” Just the viral RNA, surely all RNA is targeted during collateral cleavage?
Author Response
Comments 1:
57 – binding
Response: Thank you for catching that. The correct term is "binding," and we have made this correction in the revised manuscript.
Comments 2:
62-63 – “non-target RNAs, resulting in a broad reduction in the overall viral RNA population.” Just the viral RNA, surely all RNA is targeted during collateral cleavage?
Response: 61-67 –The sentence in question has been revised for clarity. The updated version reads: “In antiviral strategies, this collateral cleavage allows Cas13a to target not only the intended viral RNA but also multiple other viral RNAs. The potency of this activity has been demonstrated against three distinct ssRNA viruses: lymphocytic choriomeningitis virus (LCMV), influenza A virus (IAV), and vesicular stomatitis virus (VSV). This broad, indiscriminate RNA degradation is crucial for rapidly reducing viral RNA levels, enhancing Cas13a's effectiveness as an antiviral tool [22].”
Reviewer 2 Report
Comments and Suggestions for Authors
The authors have improved the manuscript significantly and have addressed majority of the raised concerns. The manuscript seems to be in a shape to be accepted with minor revisions (as detailed below) for publication in "Viruses":
1) The english in the newer version in line 57 needs to be corrected - "The bind of crRNA with its complementary RNA target ...".
2) The authors are encouraged to provide the name of the reference gene used in their reverse transcriptase real time PCR assay in Figure S3C.
3) In figure 4A, the authors provide pictures of lines L1 and L2, however, in the gel pictures (4B) they provide data for Lines 2 and Lines 6. This can get confusing to the readers. The authors are encouraged to provide pictures of L2 and L6 instead of L1 and L2 in Figure 4A, although they have shown all the Lines in Figure S3A.
Comments on the Quality of English LanguageAs mentioned above in the comment section, a minor editing is required.
Author Response
Comments 1:
The english in the newer version in line 57 needs to be corrected - "The bind of crRNA with its complementary RNA target ...".
Response: Thank you for catching that. The correct term is "binding," and we have made this correction in the revised manuscript.
Comments 2:
The authors are encouraged to provide the name of the reference gene used in their reverse transcriptase real time PCR assay in Figure S3C.
Response: Thank you for your suggestion. The reference gene used in our reverse transcriptase real-time PCR assay for Figure S3C is PP2A. We have included this information in the revised manuscript (Line 150-151) and legend for Figure S3 (Line 8-9).
Comments 3:
In figure 4A, the authors provide pictures of lines L1 and L2, however, in the gel pictures (4B) they provide data for Lines 2 and Lines 6. This can get confusing to the readers. The authors are encouraged to provide pictures of L2 and L6 instead of L1 and L2 in Figure 4A, although they have shown all the Lines in Figure S3A
Response: Thank you for pointing out the discrepancy in Figure 4A. We apologize for the confusion caused by the typo. The label "L1" in Figure 4A was mistakenly written instead of "L6." This typographical error was corrected in the revised manuscript.